# New Digital Infrastructure’s Impact on Agricultural Eco-Efficiency Improvement: Influence Mechanism and Empirical Test—Evidence from China

**DOI:** 10.3390/ijerph20043552

**Published:** 2023-02-17

**Authors:** Jin Ren, Xinrui Chen, Tingting Gao, Hao Chen, Lefeng Shi, Ming Shi

**Affiliations:** 1School of Economics and Management, Chongqing Normal University, Chongqing 401331, China; 2National Center for Applied Mathematics in Chongqing, Chongqing 401331, China

**Keywords:** new digital infrastructure, agricultural eco-efficiency, innovation infrastructure, urbanization level

## Abstract

This paper attempts to explore the overall impact of its rural digitization process on agricultural carbon emissions and non-point source pollution in the context of China. By doing so, we analyze whether digitization has an impact on agricultural pollution reduction, analyze its conductive mechanism, and draw its policy implications. To this end, the paper innovatively incorporates new digital infrastructure and urbanization level into of the concept of agricultural eco-efficiency (AEE) and adopts the SBM-DEA model, entropy weighting method, and mixed regression to analyze, based on the sample data of the 30 provinces of China from 2011 to 2020. The results indicate that: (1) new digital infrastructure has a significant contribution to the improvement of AEE of China; (2) both information infrastructure and integration infrastructure have a significant positive effect on AEE, and the effect of information infrastructure is more effective, but there is an inverted “U”-shaped relationship between innovation infrastructure and AEE level; (3) the moderating effect mechanism suggests that the level of urbanization reinforces the contribution of new digital infrastructure to AEE; and (4) the heterogeneity test shows that the effect of new digital infrastructure on AEE is more significant in regions with well-developed traditional transportation facilities and in periods when the government pays more attention to agricultural ecological issues. The above results also provide rich insights for China and other similar developing countries on how to balance the agriculture digitization and AEE.

## 1. Introduction

As the basic industry for human survival and development, the sustainable development of agriculture has always been a concern of public health and often progresses along with the development of human science and technology. However, there are always two sides of the coin. The widespread use of agricultural petroleum machinery, pesticides, and fertilizers, albeit increases the agriculture output, and also brings carbon emissions, soil poisoning, and other non-point source pollution in agriculture, which not only threatens the quality and safety of agricultural products, but also has widespread impacts on environmental protection and public health [1,2,3,4,5,6,7,8,9,10,11,12]. The above experience warns us that it is necessary to take a serious look at every technological change that takes place in agriculture, analyze its advantages and disadvantages in advance, and try to build on its strengths and avoid its negative influence [13,14]. Currently, we are in an era of digitization. In many countries and regions, the tentacles of digitization have gradually reached out to many aspects of agriculture, such as online sales of agricultural products and agricultural management through big data technology [15,16,17]. However, would digitization bring as a negative impact on the sustainable development of agriculture as other technologies? This remains an important open question.

To the above question, there has not been a specific response in existing research, though some results of other research show that digitization might bring some active effects in terms of environmental protection to other industries. For example, some scholars have found that information infrastructure construction could reduce industrial pollution emission intensity by expanding economic scale and promoting technological progress [18]. In terms of environmental benefits, the report jointly published by the international environmental protection organization Greenpeace and Renmin University of China [19] points out that the CO_2_ emissions from the production process of new digital infrastructure in China has been reduced by 13.483 million tons in 2020 compared to traditional infrastructure, with an emission reduction rate of about 7.24%. From the perspective of conductive mechanism, the above positive effects are mainly attributed to the efficiency improvement of industrial operations and the innovation of management models in their digitization process. [18,20,21,22]. Whether these effects still are effective in sustainable agriculture is the focus of this paper.

To comprehensively measure the impact of digitization on agricultural sustainability, this paper uses the application of new digital infrastructure to characterize the agriculture digitization and takes China as a case (The reasons behind choosing China are as follows: (1) China supports 20 percent of the world’s population on 9 percent of the world’s land; (2) China is currently in the process of digitizing its agriculture across new digital infrastructure construction), calculating the agriculture eco-efficiency (AEE) during the process of digitization through data collection and econometric analysis, in order to explore the impact and conductive mechanism of digitization on agricultural environmental protection. Compared with traditional agriculture, AEE emphasizes the unity of economic and environmental benefits, which not only can provide quantifiable criteria for green agricultural development, but also provide a guidance for sustainable agricultural development [8,9]. How to improve the level of AEE? The existing literature focuses on the impacts of climate change on agroecosystems and their adaptive management [4,5], agroecosystem service trade-offs and synergies [6,7], and other social and economic factors such as social structure (e.g., rural labor shift, population aging), production characteristics (e.g., agricultural scale and agricultural machinery density, industrial integration), and government behavior (e.g., agricultural fiscal ecology subsidies) [8,9,10,11,12]. However, in the background of the booming global digital economy, few studies have examined the impact of digital technologies on AEE. Although the role of technology progress on AEE has been mentioned in some of the literature, but these studies are mostly conducted based on the premise of linear assumptions. However, in fact, with reference to the environmental Kuznets curve, the impact of digital technological progress in production patterns on AEE may also be non-linear.

In the following sectors, the paper will incorporate new digital infrastructure and urbanization level into the concept of AEE and adopt DEA-SBM model, entropy weighting method, and mixed regression to analyze, based on the sample data of the 30 provinces of China from 2011 to 2020. Compared with extant studies, this paper may have the following contribution: It maybe the first study to measure the influence and the conductive mechanism of the agricultural digitization on the AEE;The corresponding indicator system and specific method are proposed and constructed innovatively;The relevant policy enlightenment is provided, based on the obtained results.

## 2. Theoretical Analysis

### 2.1. Ecological Effects of New Digital Infrastructure

Compared with traditional infrastructure, new digital infrastructure is the construction of information digitization, which is a new generation of information technology represented by digital knowledge and information as its main object, focusing on 5G, industrial internet, artificial intelligence and internet of things, which can provide digital transformation, intelligent upgrading and innovation for agriculture. So, we prefer the application of new digital infrastructure to characterize the digitization of agriculture. However, existing studies have more often discussed the role and significance of new digital infrastructure in optimizing resource allocation and promoting high-quality economic development from the macro level [20,21]; or empirically tested the role of new digital infrastructure on digital transformation in industry [22], and the induced effect of new digital infrastructure in promoting green and sustainable industrial development from the meso level [18]. In conclusion, the application of new digital infrastructure in industry is abundant, but its application in agriculture is rare, especially focusing on the ecological and environmental benefits of new digital infrastructure. Rare studies discuss new digital infrastructure and AEE in a unified framework. The existing literature focuses more on the effects of general infrastructure aspects such as the level of urbanization [23], the machinability of farmland [24], and the scale operations of land [25] on AEE. Even if a few of them involve the impact of infrastructure development on AEE, they only analyze factors such as smart agriculture [26] and information technology [27] from the theoretical level, but neglect to verify the effect and mechanism of new digital infrastructure on AEE from the empirical level.

The impact of new digital infrastructure on AEE can be realized in two aspects, as follows:

(1) The resource integration effect of new digital infrastructure. Agricultural production may suffer the risk of misallocation of resources [28], which not only causes loss of agricultural output but also leads to a decline in AEE [29]. In order to break down barriers between agriculture and other industries, new digital infrastructures can facilitate the overall articulation of various production processes, speed up the circulation of production factors by integrating and developing existing resources, realize the recombination of production factors, and use this to realize digitization, informationalization, and intelligence for the extension and restructuring of the agricultural industry chain and the green economy [30]. In the process of resource reorganization, new digital infrastructure is utilized as a carrier to agriculture digitization, and the application of a number of green technologies and processes that promote resource conservation, reduce pollution emissions, and efficient energy storage facilities has given rise to new eco-friendly agricultural production methods, providing new paths and kinetic energy to enhance AEE. In addition, resource reorganization and integration will generate scale effects, which itself has positive externalities of energy saving and emission reduction [31]. Scale farming has been proven to be beneficial for increasing soil organic matter, boosting agricultural land fertility, lowering the need for chemical fertilizers, and enhancing the soil environment [32,33].

(2) Technological progress effect of new digital infrastructure. First, green technological innovation with new digital infrastructure has enabled changes in traditional agricultural production mode that are more environmentally friendly and with positive AEE. For example, Yangling Smart Agricultural Demonstration Park in Shaanxi, China, relies on Internet of Things technology to establish a real-time collection system for “temperature, light, air, water and fertilizer” information in various types of greenhouses, which allows real-time monitoring and effective control of agricultural non-point source pollution and emissions. Second, the new digital infrastructure has encouraged the development and application of agricultural information technology in the storage, marketing, and development of agricultural products. For example, the establishment of agricultural products e-commerce platform helps to expand the space and channels of agricultural products marketing and drive the emergence of new models such as digital farms. In addition, the biased choice of green agricultural products will inevitably force agricultural resource consumption to shift in the direction of energy saving and carbon reduction, reduce agricultural pollutant emissions, effectively achieve ecological environmental protection, and promote the ecological construction of agricultural infrastructure. With the supply chain system, detailed information on the origin of agricultural products can be provided to customers, then tracking and monitoring the safety of agricultural products and the protection of public health can be realized. Thirdly, with the release of information platforms and the convenient infrastructure, rural resources can be matched with public leisure tourism and health care, while also effectively improving public health and enhancing the livable environment in rural areas. In conclusion, green technology innovation with digital new infrastructure expands the agricultural value chain, enhances agricultural production efficiency, reduces agricultural pollutant emissions and improves the rural living environment [34]. Based on this, this paper proposes the following hypotheses:

**Hypothesis** **1.***There is a significant positive impact of new digital infrastructure on AEE*.

### 2.2. The Threshold Effect of Innovative Infrastructure

According to the definition of China’s National Development and Reform Commission, the new digital infrastructure contains three components: information infrastructure, convergence infrastructure, and innovation infrastructure. Among them, innovation infrastructure refers to the infrastructure with public welfare attributes that supports scientific research, technology progress, and product development. Technological innovation is the source of eco-efficient development [35], but there are many uncertainties in the process of agroecological construction, which can be influenced by factors such as labor, policy and technology. Especially in the current context of increasingly severe resource and environmental constraints, innovation cannot simply pursue the increase in output, but should also balance the development of natural and social elements. Some scholars have found that the impact of technology innovation on AEE may be nonlinear [36], so there is variability in the impact of innovation infrastructure on AEE at different levels of innovation. In the early stage of technology innovation, agricultural production relied more on inputs of production factors and to some extent ignored environmental protection issues. At this stage, technological innovation not only improves the production efficiency, promotes agricultural output and provides technical support for environmental protection, but the intensive effect of production also reduces resource consumption and alleviates ecological pressure [37]. While when technological innovation develops to a certain level, technology innovation also derives new environmental pollution problems. For example, the use of agricultural machinery, although it improves agricultural production efficiency, it also increases the energy consumption and agricultural carbon emissions, leading to the incompatibility between agricultural development and environmental protection. Thus, in this progress, the technological innovation on AEE may suffer diminishing marginal utility. Moreover, in the context of strengthened resource constraints, there may be mutual resource crowding and uneven development between digital technology innovation and AEE improvement. Therefore, whether there is a stage-dependent impact relationship between innovation infrastructure and AEE deserves further discussion. Based on the above analysis, hypothesis 2 is proposed as follow:

**Hypothesis 2.** 
*There is variability in the impact of innovation infrastructure on AEE at different levels of innovation. When the level of innovation is within a certain threshold, innovation infrastructure has a positive effect on AEE.*


### 2.3. Moderating Effect of Urbanization Level

AEE, as an integrated expression of human agricultural production activities, is affected by the level of regional economic development. The promotion of urbanization has led to economic development and ecological civilization construction [38,39]. In the progress of urbanization, the rural population shifts to the cities, and this process itself directly reduces the direct pollution of agricultural ecological environment by human activities. The population shift caused by urbanization can provide labor resources for urban development; the economic development and technology diffusion in the urbanization process are in turn conducive to the transformation of agricultural operations from rough to large-scale, which provides better environmental and production factor resources for agroecological development and reduces agricultural carbon emissions [40,41]. At the same time, urbanization reinforces the spread of green agricultural consumerism and the demand for green agricultural products, therefore farmer households will pay more attention to green agricultural production, increase green inputs, reduce the use of chemical substances, and improve AEE. The urbanization also provides more convenient transportation facilities for storage and logistics of agricultural products, promoting the green development of agriculture [23]. Therefore, in the mechanism of new digital infrastructure boosting AEE growth (Figure 1), a higher level of urbanization can break the dilemma of inadequate resource flow and push forward the implementation of various infrastructures, so that the resource integration effect and technological progress effect of new digital infrastructure can be given full play, which in turn promotes AEE growth [42]. Based on the above analysis, Hypothesis 3 is proposed as follow: 

**Hypothesis 3.** 
*The level of urbanization plays a positive moderating role in the impact of new digital infrastructure on AEE.*


## 3. Models, Estimation Methods, and Variables

### 3.1. Econometric Models and Estimation Methods

In order to investigate the impact of new digital infrastructure on AEE, this paper establishes a mixed regression model for analysis, which gives the best parameter fit in statistical significance by minimizing the squared error and finding the best functional match of the data, with the specific model set as Equation (1). This paper discusses the effects of the composite index of new digital infrastructure and the three sub-dimensions on AEE under variable and constant returns to scale, respectively.
(1)$${\mathrm{AEE}}_{\mathrm{it}}=\alpha_{0}+\alpha_{1}{\mathrm{NDI}}_{\mathrm{it}}+\alpha_{2}{\mathrm{Control}}_{\mathrm{it}}+\varepsilon_{\mathrm{it}}$$

In Equation (1), i denotes the region. t denotes the year. ${\mathrm{AEE}}_{\mathrm{it}}$ is the rural eco-efficiency with variable and constant returns to scale in period t of province i. ${\mathrm{NDI}}_{\mathrm{it}}$ is the level of new digital infrastructure in period t of province i. ${\mathrm{Control}}_{\mathrm{it}}$ denotes the control variable containing the level of agricultural economic development (AED), cropping structure (PS), agricultural disaster rate (ADR), rural human capital (RHC), and the level of financial support to agriculture (FSA). $\varepsilon_{\mathrm{it}}$ is the random error term.

According to the previous theoretical analysis, there may be variability in the impact of innovation infrastructure on AEE due to different levels of innovation. Therefore, to further explore the variability of impact, the explanatory variable innovation infrastructure (INNI) itself was selected as the threshold variable to build the threshold model.
(2)$${\mathrm{AEE}}_{\mathrm{it}}=\beta_{0}+\beta_{1}{\mathrm{INNI}}_{\mathrm{it}}\times I\left({\mathrm{INNI}}_{\mathrm{it}}\leq \theta \right)+\beta_{2}\mathrm{INNI}\times I({\mathrm{INNI}}_{\mathrm{it}}>\theta )+\beta_{3}{\mathrm{Control}}_{\mathrm{it}}+\varepsilon_{\mathrm{it}}$$

In Equation (2), ${\mathrm{INNI}}_{\mathrm{it}}$ is the threshold variable for the period t Province i. I(·) is the indicator function of the threshold variable. $\beta_{i}$ is the variable coefficient. θ is the variable threshold. $\varepsilon_{\mathrm{it}}$ is the random error term.

In addition, this paper includes urbanization levels in the analysis framework to further examine the moderating mechanism of urbanization levels at different levels in the process of new digital infrastructure affecting AEE. The interaction term between new digital infrastructure and urbanization level is added to Equation (1) to obtain Equation (3).
(3)$${\mathrm{AEE}}_{\mathrm{it}}=\alpha_{0}+\alpha_{1}{\mathrm{NDI}}_{\mathrm{it}}+\alpha_{2}{\mathrm{LU}}_{\mathrm{it}}+\alpha_{3}{\mathrm{NDI}}_{\mathrm{it}}\times {\mathrm{LU}}_{\mathrm{it}}+\alpha_{4}{\mathrm{Control}}_{\mathrm{it}}+\varepsilon_{\mathrm{it}}$$

In Equation (3), ${\mathrm{LU}}_{\mathrm{it}}$ denotes the urbanization level of province i in period t, and $\varepsilon_{\mathrm{it}}$ is the random error term.

### 3.2. Variables

#### 3.2.1. Explained Variable

Level of agricultural eco-efficiency (AEE). There are more quantitative research methods for measuring the level of AEE in existing studies, such as DEA measures or stochastic frontier methods. Among them, Charnes et al. [43] first proposed a DEA analysis method based on the condition of constant returns to scale; Banker et al. [44] changed the premise of constant returns to scale to variable returns to scale to study the relative levels between different sectors; and in recent years, the academic community favors the SBM-DEA model containing non-desired output to measure the level of AEE, but less decision units with AEE levels equal to or exceeding one have been included in the study for discussion. Therefore, in this paper, we choose the SBM-DEA model that considers constant scale returns and variable super-efficiency to measure the AEE level, and the specific model is:(4)$$\mathrm{MinAEE}=\frac{\frac{1}{m}\sum_{i=1}^{m}\left(\frac{\;\bar{x}}{x_{\mathrm{ik}}}\right)}{\frac{1}{r_{1}+r_{2}}\left(\sum_{s=1}^{r_{1}}\left(\frac{\bar{y^{d}}}{y_{\mathrm{sk}}^{d}}\right)+\sum_{q=1}^{r_{2}}\left(\frac{\bar{y^{u}}}{y_{\mathrm{qk}}^{u}}\right)\right)}$$
(5)$$\left\{ \begin{array}{c} \bar{x}\geq \sum_{j=1,\neq k}^{n}x_{ij}\lambda_{j};\bar{y^{d}}\leq \sum_{j=1,\neq k}^{n}y_{sj}^{d}\lambda_{j};\bar{y^{d}}\geq \sum_{j=1,\neq k}^{n}y_{qj}^{d}\lambda_{j}\\ \bar{x}\geq x_{k};\bar{y^{d}}\leq y_{k}^{d};\bar{y^{u}}\leq y_{k}^{u}\\ \lambda_{j}\geq 0,i=1,2,\ldots ,r_{2}\\ s=1,2,\ldots r_{1};q=1,2,\ldots r_{2} \end{array}\right.$$

In Equations (4) and (5), n denotes the number of decision units. m denotes the number of input indicators. $r_{1}$ denotes the number of desired output indicators. $r_{2}$ denotes the number of non-desired output indicators. $x,{\;y}^{d},{\;\mathrm{and}\;y}^{u}$ denote the elements of the input matrix, desired output matrix, and non-desired output matrix, respectively.

In terms of quantifying the level of AEE, the correspondence between input and output indicators in the existing literature needs to be more clearly defined, and in terms of the selection of non-desired output indicators, most scholars base their research on agricultural non-point source pollution or carbon emissions, and relatively few research results combine the two. The specific descriptions of the selection of each indicator in this paper are as follows.

(1) Input indicators. In this paper, four aspects such as agricultural natural resources, labor, machinery, and chemical inputs are selected as input indicators. ① Natural resource inputs. There is a rich variety of natural resources. However, the main ones that can specifically represent agricultural production are land and water resources, and crops cannot be grown without soil and water. So, this paper uses the total area of sown crops and agricultural irrigation area to represent land and water resources of natural resources, respectively [9], unit (thousand hectares). ② Labor input. From the development history of Chinese agriculture, although the use of mechanized equipment has been gradually introduced in agricultural production, labor input is still inseparable. In this paper, the total number of agricultural employees is chosen as a proxy variable for labor input, and considering that there is no directly engraved number of agricultural employees, the number of agricultural employees = the number of people employed in the primary industry × (agricultural output value/total output value of agriculture, forestry, animal husbandry and fishery, etc.) is chosen to be re-engraved [45], unit (10,000 people). ③ Machinery input. With the progress of agricultural science and technology, agricultural production is moving toward scale and mechanization, and the process of agricultural modernization is accelerating, prompting the input of a large number of agricultural machinery, and this paper selects the total power of agricultural machinery and the amount of agricultural diesel fuel used as the proxy variables of mechanical input [46], units (million kilowatts) (million tons). ④ Chemical input. In agricultural production, chemical substances such as pesticides, fertilizers, and agricultural films are inevitably used, so this paper reflects the proxy variables of chemical inputs as a composite indicator consisting of pesticide use + fertilizer refraction + agricultural film use [46], unit (million tons).

(2) Expected indicators. The desired output of agriculture contains two categories: the level of agricultural development and the level of agroecology, where the level of agricultural development is replaced by the total agricultural output value, unit (billion yuan).

The agroecological level is expressed by carbon sequestration in agriculture (CSA), and only the carbon sequestration of major crops during the complete life cycle of the production process considered [47], unit (million tons). In addition, its accounting Equation (6) is as follows.
(6)$$\mathrm{CSA}=\sum_{i=1}^{15}{\mathrm{CSA}}_{i}=\sum_{i=1}^{15}\frac{\left(1-r\right){\mathrm{cs}}_{i}y_{i}}{{\mathrm{HI}}_{i}}$$

In Equation (6), CSA is the total crop carbon uptake of agricultural production activities, ${\mathrm{CSA}}_{i}$ represents the uptake of each type of crop, i represents the i’s crop, r represents the water content of the crop, ${\mathrm{cs}}_{i}$ represents the emission coefficient of each carbon emission source, $y_{i}$ represents the yield of the i’s crop, and ${\mathrm{HI}}_{i}$ represents the economic coefficient of the crop, which is determined based on the following Table 1 [48,49].

(3) Non-desired output indicators. Agricultural non-desired output contains two categories of carbon emissions and pollution emissions [50], and this paper selects carbon emissions and agricultural non-point source pollution as non-desired indicators based on the availability of data. Among them, agricultural carbon emissions are reflected in agricultural carbon emissions caused by six factors such as pesticides, fertilizers, diesel, agricultural films, irrigation, and tillage [51], so this study uses agricultural carbon emissions as a proxy variable for non-desired output, unit (million tons), and its accounting formula is as follows [51,52,53]:(7)$$\mathrm{CE}=\sum_{i=1}^{6}{\mathrm{CE}}_{i}=\sum_{i=1}^{6}\beta_{i}A_{i}$$

In Equation (7), CE is the total carbon emissions from agricultural production activities, ${\mathrm{CE}}_{i}$ represents the emissions from various carbon sources, i represents the i’s carbon source, $A_{i}$ represents the original amount of the i’s carbon emission source, and $\beta_{i}$ represents the emission coefficient of each carbon emission source, which is determined based on the following Table 2 [52,53].

In addition, the problem of agricultural non-point source pollution has always existed. Due to the use of chemical substances such as pesticides and fertilizers that exacerbate soil and water pollution, agricultural non-point source pollution threatens the quality and safety of agricultural products and is a constraint that affects the sustainable development of agriculture. Therefore, in order to reflect the problem of pollution sources in agriculture such as planting, livestock breeding and aquaculture, this paper selects four components consisting of chemical oxygen demand (COD), ammonia nitrogen (NH), total nitrogen (TN), and total phosphorus (TP) emissions summed up as a proxy variable for agricultural non-point source pollution [54], unit (million tons).

The results measured under variable and constant returns to scale are shown in Figure 2 (due to space limitations, only data changes at the beginning of the period 2011, the middle of the period 2016 and the end of the period 2020 are reported).

From Figure 2, it can be seen that the level of AEE in each province shows an increasing trend in 2011, 2016, and 2020, both in the case of variable scale returns and constant scale returns. AEE varies substantially between Chinese provinces, and overall, the eastern part of the country has far higher levels of AEE than the western part. The highest level of AEE is in Beijing, and the lowest is in Gansu. Beijing is the capital of China, a mega-city with a population of more than 21.88 million, and a highland for advanced technologies, excellent talents, and innovative resources. The penetration of digital technologies of modern agriculture in Beijing has resulted in an urban modern agricultural ecological service value of 400 billion RMB [55]. The main ecological environment problems in Gansu are land desertification, frequent sandstorm weather, water shortage, and poor environmental self-cleaning capacity. Its agricultural industry features traditional cold and dry agriculture. On the other hand, due to the geographical location and economic level, similar to Gansu, the foundation and process of digital infrastructure and digital agriculture in northwest China is slower than developed areas in the east. Therefore, the western region develops special green ecological agriculture by digital transformation, integrated application of modern facilities and advanced agricultural technologies, and combining with its own resource endowment. As seen from Figure 2, the regions with developed economy and better digitization have higher AEE, while the regions with poor economy and weaker digitization foundation have lower AEE. This also preliminarily indication of the role of agricultural digitization in promoting AEE. 

#### 3.2.2. Explanatory Variable

New digital infrastructure (NDI). Compared with traditional infrastructure, new digital infrastructure is essentially the infrastructure construction of information digitization, which is a new generation of information technology represented by digital knowledge and information as its main object of action, focusing on 5G, industrial Internet, artificial intelligence, and Internet of Things, which can provide digital transformation, intelligent upgrading, and integration and innovation services for various industries. In 2020, China’s National Development and Reform Commission proposed the development of new digital infrastructure to cover three major areas of information, convergence, and innovation, and give full play to its role in promoting high-quality economic development. Therefore, this paper constructs the new digital infrastructure development level index and index system (Table 3) based on the above three fields and under the principles of ensuring data availability, systematic science, objectivity and comprehensiveness, and hierarchy. The specific measurement steps of the comprehensive evaluation index of new digital infrastructure development level are as follows: First, the entropy weight method is chosen to measure the development indexes of information infrastructure and innovation infrastructure, respectively; second, according to the data availability, there is no direct index ratio representation in convergence infrastructure at present, meanwhile, convergence infrastructure is an optimization and upgrading on traditional infrastructure, which reflects the degree of information and digital integration in the field of traditional infrastructure. Therefore, this paper is based on the coupled coordination model to measure the coupled coordination degree of both traditional infrastructure and informatization degree to portray the development level of convergence infrastructure; thirdly, the three sub-dimensional indicators of new digital infrastructure are measured by entropy weight method, then for the synthesis of comprehensive indicators, the secondary measurement cannot be simply performed by entropy weight method, considering that the dynamic process of new digital infrastructure development will be influenced by national policies, industry experts and various technical funds Considering that the dynamic process of new digital infrastructure development will be influenced by national policies, industry experts and various types of technical funds, it is necessary to give preference to three sub-dimensions in order to evaluate new digital infrastructure development scientifically, realistically and operationally. In this paper, the comprehensive indicators of new digital infrastructure are measured based on the hierarchical analysis method. In order to simplify the operation procedure and reduce the interference of human factors, the three indicators of information infrastructure, convergence infrastructure and innovation infrastructure are given weights by selecting experts’ scoring method, in which there are five experts from government, industry, and university (the number of personnel is 1, 2, and 2 in order), and the final weights are 0.4, 0.3, and 0.3, respectively, the corresponding index values were derived through hierarchical analysis, and finally the comprehensive index values was calculated by weighting, and the specific measurement results are shown in Table 4 (due to space limitation, only the dynamic changes in 2011 and 2020 are shown).

Table 4 shows that the overall level of new digital infrastructure development in the eastern, southern, and coastal regions is higher, while the further away from the coastal regions, the lower the overall level of new digital infrastructure development. The reason for this situation is that the eastern, southern, and coastal regions have long had a clear advantage in terms of economic development dynamics, infrastructure development, and business environment, and despite the implementation of the Western Development Strategy and the Rise of Central China Strategy for the inland regions, which have promoted regional economic development, the gap between the inland regions and the eastern, southern, and coastal regions has tended to widen further.

#### 3.2.3. Moderating Variables

Level of urbanization (LU): Urbanization is an important carrier of ecological civilization construction, which can play a suppressive role on agricultural carbon emissions and promote green development of agriculture. At the same time, the development of urbanization in the region will be accompanied by the investment of various types of infrastructure, improving the existing regional economic development environment, promoting the scale of agricultural production, and improving AEE [41]. Therefore, the level of urbanization in this paper is expressed by the urbanization rate, which is the proportion of the urban population to the total population in each region at the end of the year, unit (%).

#### 3.2.4. Control Variables

To minimize the bias caused by omitted variables, according to the existing literature [56,57,58], the following control variables are selected in this paper: (1) the level of agricultural economic development (AED), expressed as the ratio of the level of total agricultural output value to the resident population in each region, unit (million yuan/person); (2) planting structure (PS), expressed as the ratio of grain sown area, unit (%) (3) agricultural disaster rate (ADR), expressed as the ratio of crop disaster area to total crop sown area, unit (%); (4) rural human capital (RHC), expressed as farmers’ education level, unit (year); and (5) financial support to agriculture (FSA), expressed as the ratio of local expenditure on agricultural and forestry affairs to general fiscal budget expenditure, unit (%).

#### 3.2.5. Data Sources and Descriptive Statistics

This study uses China as an example to illustrate the role of new digital infrastructure on AEE. The study sample is a panel of 30 provinces, municipalities directly under the Central Government and autonomous regions of China excluding Tibet and Hong Kong, Macao and Taiwan from 2011 to 2020, and the data were obtained from the China Statistical Yearbook, China Rural Statistical Yearbook and statistical yearbooks of provinces, municipalities directly under the Central Government and autonomous regions in previous years. The final 1% tailing process was performed. The descriptive statistics of each variable are shown in Table 5.

## 4. Empirical Testing

### 4.1. Baseline Regression

In this paper, a mixed regression model is used for analysis to estimate the parameters of Equation (1), and the estimation results are shown in Table 6. Where Columns (1)–(4) are the estimated results of variable returns to scale on AEE, and Columns (5)–(8) are the estimated results of constant returns to scale on AEE. Columns (1) and (5) in Table 6 show the results of the baseline regression. According to the results of the baseline regression, the influencing factors of new digital infrastructure (NDI) are all significantly positive at the 5% level, indicating that the development of new digital infrastructure helps to improve the level of AEE, and each 1 unit increase in new digital infrastructure will promote 1.0237 units of AEE under the case of variable returns of scale (VRS-AEE); under the case of constant returns of scale (CRS-AEE), each 1 unit increase in the case of constant scale returns, every 1 unit increase in new digital infrastructure will promote 0.1917 unit increase in AEE.

Since the new digital infrastructure is synthesized from the three subindexes of information infrastructure, convergence infrastructure, and innovation infrastructure, in order to further analyze which specific dimensions of new digital infrastructure affect AEE, this study chooses a mixed regression model to verify the results of the three sub-dimensions of new digital infrastructure on AEE (Table 6). Columns (2)–(4) show the regression results for information infrastructure, integration infrastructure, and innovation infrastructure with variable returns to scale; Columns (6)–(8) show the regression results for information infrastructure, integration infrastructure, and innovation infrastructure with constant returns to scale. The results show that the increase in information infrastructure and integration infrastructure is beneficial to the increase in AEE under both variable and constant returns to scale. In terms of impact coefficients, the information infrastructure has the largest impact coefficient, followed by the convergence infrastructure. With the continuous improvement of rural digital infrastructure and upgrading of traditional infrastructure, from the supply side, digital technology can help farmers reduce production losses brought on by factor distortions, increase the effectiveness of business decisions, optimize the allocation of production factors, and reduce the frequency and dosage of pesticide application, thereby reducing agricultural carbon emissions and improving the safety of agriculture. From the demand side, consumers can rely on the big data platform to track the green production, clean storage, and safe transportation of agricultural products in real time, enhancing farmers’ external motivation to develop green and low-carbon agriculture. All is made possible by the integration and transformation of information infrastructure and traditional infrastructure, which offers a fresh perspective to improve the ecological effectiveness of agriculture. Therefore, digital agriculture is a crucial endeavor for developing nations to overcome the issue of small-scale and extensive agricultural production and progress toward intensification and ecological development. 

### 4.2. Threshold Effect of Innovation Infrastructure

As shown in Columns (4) and (8) in Table 6, for the innovation infrastructure sub-dimension, the coefficient of AEE under constant scale returns is significant and negative at the 1% level, while the coefficient of AEE with variable returns to scale is not significant. Innovation infrastructure is an infrastructure with public welfare attributes to support scientific research, technology development, and product development, which generally has huge investment, long investment return cycle, and uncertainty. Therefore, in order to further investigate whether there is a threshold between innovation infrastructure and AEE with variable returns to scale, this paper selects the threshold model for analysis. Firstly, we determine whether there is a threshold effect, and if so, we need to further test the threshold value and determine how many thresholds exist and choose which type of threshold model to use. Since AEE with constant returns to scale has met significance, only the effect of innovation infrastructure on AEE with variable returns to scale is considered. As seen in Table 7, a single threshold model should be chosen for this paper, where the threshold value is 0.2813.

From the results of the threshold regression in Table 8, when the innovation infrastructure is below 0.2813, the innovation infrastructure on VRS-AEE is significantly positive at the 1% level, the innovation infrastructure plays a positive role in promoting AEE, and for each unit of innovation infrastructure increase, the AEE increases by 14.9609 units, which means that along with the transformation of traditional infrastructure innovation, strengthening the construction of innovation infrastructure will promote the growth of AEE, while when innovation infrastructure crosses the threshold value of 0.2813, innovation infrastructure on VRS-AEE is significantly negative at the 5% level, and innovation infrastructure has a negative inhibitory effect on AEE, with AEE decreasing by 4.1182 units for each unit of innovation infrastructure increase. Therefore, innovation infrastructure has an inverted U-shaped characteristic of promoting and then inhibiting the development of AEE, which reflects to a certain extent that innovation is the first driving force of ecological civilization development, but attention should also be paid to the coordinated development of innovation and ecological environment. When innovation development started to take off, in regions that mainly rely on labor and other factors of production for production activities, in order to make up for technological shortcomings, the investment of human and financial resources in agriculture was increased, which led to an increase in the proportion of knowledge-intensive industries, which not only promoted the output of more products and services, but also reduced resource consumption, eased ecological pressure, and provided technical support for environmental protection [59]. However, under the premise of resource constraint, innovation infrastructure still has the natural monopoly property of traditional infrastructure, innovation infrastructure investment may crowd out resources for ecological construction, and there is a high degree of uncertainty whether R&D innovation can be recognized by the market and generate positive environmental benefits. Therefore, the relationship between innovation infrastructure and AEE is non-linear at different stages, showing an inverted U-shaped characteristic of first promoting and then inhibiting.

### 4.3. Robustness Tests

#### 4.3.1. One-Period Lag Analysis

Considering that there may be a time lag effect on the impact of new digital infrastructure on AEE, in order to further test hypothesis 1 and to verify whether there is a causal inversion problem in the mixed regression model, the data of the core explanatory variables with one period lag were included in the model and the mixed regression analysis was re-run, and the results of the robustness test are shown in column (1) and (2) of Table 9. The results also coincide with the regression results in Table 6, which proves that the results of this paper are robust.

#### 4.3.2. Replacement of the Baseline Model Regression Method

In the previous paper, the mixed regression model was chosen to measure the impact of new digital infrastructure on AEE. In order to avoid the bias brought by the method selection and to ensure the credibility and stability of the estimation results, the Tobit model [60] was chosen to re-measure in this paper. Its general form is as follows.
(8)$$\{\begin{array}{c} y^{*}=\beta X_{i}+\mu_{i}\\ y_{i}=y_{i}^{*}\\ y_{i}=0 \end{array}$$

In Equation (8), $y_{i}^{*}$ is the latent variable. $y_{i}$ is the observed dependent variable. $X_{i}$ is the vector of independent variables. β is the vector of correlation coefficients, and the error term $\mu_{i}$ is independent and follows a normal distribution.

Since the semi-parametric estimation method does not need to assume the specific form of the residuals, consistent estimates can be obtained even in the case of individual heteroscedasticity, and at the same time, considering the specificity of the research content of this paper, the fixed-effects Tobit model cannot find sufficient statistics of individual heterogeneity, so the fixed-effects Tobit model is not considered, and the mixed Tobit method is used in this paper. The regression of the model, the results of which are shown in Columns (3) and (4) of Table 9, shows that the impact coefficient of new digital infrastructure significantly promotes the growth of AEE at the 5% level for both variable and constant scale returns, again indicating the robustness of the research findings.

### 4.4. Heterogeneity Tests

The robustness of the baseline results has been demonstrated by the above estimation, and in order to avoid problems such as reverse causality and variable measurement errors that affect the accuracy of the results, this paper uses the instrumental variables method for 2SLS regression to solve the problem of endogeneity that exists. For the selection of instrumental variables, this paper refers to the research methods of Hu and Zhong, and Zhao [10,61] to construct the interaction term between the number of fixed telephones per 100 people in 1984 and the Internet broadband access ports in that year as the instrumental variable of new digital infrastructure. The main reasons are: first, the fixed telephone penetration rate represents the construction and application basis of local communication infrastructure in the context of traditional infrastructure, and to some extent represents the predecessor of digital infrastructure, which meets the relevance requirements of the instrumental variable; second, fixed telephones do not have a direct impact on AEE, and the 1984 data are also far removed from the sample data in this paper, and have almost no time. Therefore, the condition of exogeneity of instrumental variables is satisfied. However, considering the metric problems in the application of the fixed-effect model, the interaction term between the number of fixed telephones and the Internet broadband access ports in that year was constructed as an instrumental variable for the new digital infrastructure.

The test results of the instrumental variables are shown in Table 10. From the regression results of the first stage, the interaction term consisting of the number of fixed telephones per 100 people in 1984 and the Internet broadband access ports in that year has a significant positive correlation on the new digital infrastructure, and it is significant at the 1% confidence level. From the test results, its corresponding F-statistic is 170.777, which is greater than the 10% threshold of significance level, indicating that there is no weak instrumental variable problem, so the validity of the instrumental variable is guaranteed. From the regression results of the second stage, the core explanatory variables NDI are all significantly positive at the 1% level. It indicates that after alleviating the endogeneity problem, the new digital infrastructure still has a significant contribution to AEE, and the conclusions of this paper remain robust.

### 4.5. Heterogeneity Tests

#### 4.5.1. Heterogeneity of Government Attention

From the perspective of international organizations, organizations such as the Food and Agriculture Organization of the United Nations (FAO) and the European Union (EU) attach great importance to sustainable agricultural development, and the FAO convened the World Agroenvironmental Congress and issued the “Damboz Declaration and Programme of Action on Sustainable Agriculture and Rural Development”, and the EU revised the “Common Agricultural Policy”. “Under the influence of international organizations, countries have also started to systematically search for sustainable agroecological development, such as “environmentally conserved agriculture” in Japan, “environmentally friendly agriculture” in Korea, and the top-down agroecological movement in Latin America. Agroecological movements in Latin America, and the effectiveness of various measures and movements have provided an important basis for the government to promote agroecological development [62].

Considering that national policy orientation is an important factor influencing China’s agricultural development, the difference in government attention may make the promotion of new digital infrastructure for AEE development heterogeneous. In 2017, China proposed from the top level to focus on solving the outstanding problems of agricultural environment and strengthen the prevention and control of agricultural carbon emissions and non-point source pollution, i.e., agriculture should be developed while focusing on green production and strengthening In 2018, China’s Ministry of Agriculture and Rural Development further strengthened financial subsidies for agricultural green development and introduced relevant initiatives for “green and high-quality agricultural development”. Therefore, in order to test the impact of new digital infrastructure on the development of AEE during different periods of government attention, this paper divides the sample years into two intervals from 2011 to 2016 and from 2017 to 2020, each of which represents a different degree of agricultural ecological development policy, with 2011 to 2016 set as the interval of low policy attention and 2017 to 2020 as the interval of high policy attention. Interval. Sub-sample regressions were conducted to reveal the heterogeneous effects of new digital infrastructure development on AEE growth under different levels of policy attention. Since the regression results for the constant scale return case are not significant, only the heterogeneous impact of new digital infrastructure on variable AEE in terms of scale return is considered, and the regression results are presented in column (1) and (2) of Table 11.

It can be found that the Impact coefficient of new digital infrastructure is significantly positive at the 10% level in the sample interval from 2011 to 2016 and at the 5% level in the sample interval from 2017 to 2020, and the impact coefficient from 2017 to 2020 is 2.6258, which is significantly higher than the impact coefficient from 2011 to 2017 of 0.2262. Thus, it shows that that the contribution of new digital infrastructure to the growth of AEE has expanded with the increase in national attention to agricultural ecological policies. This also confirms, to some extent, that the state’s policy support for agroecology has an important guiding and guaranteeing role, which effectively promotes the growth of green total factor productivity in agriculture.

#### 4.5.2. Heterogeneity in the Degree of Developed Traditional Transportation Facilities

Whether in developed or developing countries, traditional infrastructure development has always been the cornerstone of new digital infrastructure development. Especially in Asian developing countries, infrastructure construction effectively supports sustainable agricultural development and is a priority area for most Asian countries in the use of fiscal funds [63,64]. Therefore, traditional transportation infrastructure has been playing an indispensable role in rural economic development. It not only effectively reduces the total cost of agricultural products, including production cost, transportation cost, storage cost, and marketing cost, and improves agricultural production efficiency; it also expands the scope of market exchange for agricultural products and increases the demand for products, thus promoting the advanced rural industrial structure and modernization of traditional agriculture. The new digital infrastructure, represented by 5G, big data, and industrial Internet technologies, has injected new and powerful kinetic energy for innovative agricultural production and development, and laid the foundation for further promoting agricultural ecological development. Then, does the role of new digital infrastructure on AEE differ in different regions with developed traditional transportation infrastructure?

To verify the above question, this paper takes the top 15 regions as the regions with developed traditional transportation facilities (Inner Mongolia, Hebei, Heilongjiang, Xinjiang, Liaoning, Shandong, Henan, Shanxi, Jilin, Shaanxi, Hunan, Guangxi, Sichuan, Hubei, and Anhui) and the bottom 15 regions as the less developed regions (Guangdong, Jiangxi, Gansu, Yunnan, Fujian, Guizhou, Jiangsu, Zhejiang, Qinghai, Chongqing, Ningxia, Beijing, Tianjin, Hainan, and Shanghai), and then conduct a subsample regression. Since the regression results are not significant for the constant scale return case, only the heterogeneous effect of new digital infrastructure on variable scale return AEE is considered, and the regression results are shown in column (3) and (4) of Table 11, where the estimated coefficient of new digital infrastructure in regions with developed traditional transportation infrastructure is 0.2753 and significant at the 5% level. This indicates that new digital infrastructures have a catalytic effect on AEE in areas with developed traditional transportation infrastructure and the effect is more significant in areas with developed traditional transportation infrastructure, implying that the construction of traditional transportation facilities should be accelerated in future development to facilitate the role of new digital infrastructure.

### 4.6. Testing the Impact Mechanism: Moderating Effect of Urbanization Level

This study set up a positive moderating effect of urbanization level in the effect of new digital infrastructure on AEE. To empirically test this theoretical hypothesis, the model was estimated using the time-fixed effects approach, and the results are shown in Table 12, with Columns (2) and (4) showing the interaction effects of urbanization level and new digital infrastructure. From the results in Table 12, when the interaction term of urbanization level and new digital infrastructure is introduced, the coefficient of new digital infrastructure in the case of AEE with constant scale return becomes unstable because the effect of new digital infrastructure on AEE changes from $\alpha_{1}$ in Equation (7) to $\left(\alpha_{1}+\alpha_{3}NDI\right)$ in Equation (10) when the interaction term of urbanization level and new digital infrastructure is added. Therefore, this paper chooses the scale return variable AEE to discuss the role of urbanization in the new digital infrastructure on AEE. The results show that the coefficient of the interaction term between urbanization level and new digital infrastructure is significantly positive at the 1% level, indicating that urbanization development plays a moderating role, i.e., urbanization strengthens the positive influence of new digital infrastructure on AEE development, and hypothesis 3 is verified.

## 5. Policy Recommendations, and Research Outlook

Based on the above findings, this paper proposes policy recommendations as follows:

(1) Sticking to the desire of green and sustainable development, governments should find a balance between agricultural digitization and ecological environmental protection, build a new ecological infrastructure system, empower agricultural green transformation through digital technology, and promote ecological protection construction. However, new digital infrastructure has public goods externalities, and “free-rider” and underinvestment is common. In particular, for developing countries, it may be difficult for farmers to effectively access and master digital farming technologies and to afford the large investments in digital infrastructure and the risks associated with those investments. However, with the spread of mobile Internet access, the application of big data and the establishment of e-commerce platforms can allow farmers to use relevant digital technologies at a marginal cost close to zero, so innovations in digital infrastructure and business models can help reduce the cost of digital technology adoption. In order to ensure that agricultural activities are based on the effective use of modern information technology, developing countries should encourage a variety of entities, in addition to the government, to participate in the development of rural digital infrastructure, help with the design of local solutions, and continuously strengthen the application of modern information technology in clean agricultural production, recycling, storage, and marketing of agricultural products through information infrastructure. 

(2) It can be expected that digital agriculture may imply a major transformation of agricultural production systems, rural economies, communities, and natural resource management. For developing countries, in order to reduce the cost of transformation, it is important to pay attention to the development of convergence infrastructure in the transformation process and to promote the sharing of resources, facilities, and space between information infrastructure and traditional infrastructure. Green transformation and upgrading of traditional agricultural infrastructure can be more resource-efficient in some areas, which will be more resource-efficient, with faster fruit and expanded reproduction. Of course, in the process of promoting the integration and innovation of old and new digital infrastructures, the new generation information technology should play a leading role to amplify the ecological function of infrastructures and realize the integration and innovative development. Therefore, to lower the cost of digital usage by agricultural operators, it is advised to establish convergent value business between the government, financial institutions, mobile network operators, device makers, and service providers.

(3) The threshold feature of innovation infrastructure requires countries to further focus on the mutual coordination of agricultural digitization progress and agricultural sustainability according to the reality of technological innovation inputs on AEE. According to the current situation and actual needs of regional agricultural development, we should reasonably regulate innovative infrastructure inputs and other production factor inputs through institutional design, and be alert to the inflection point of declining AEE. Developing countries can learn from the experiences of the European Union and Japan to incentivize low-cost inclusive technological innovation. In order to encourage the friendliness of agricultural digitization, for instance, the European Union has created networks for knowledge sharing with public and private stakeholders. In addition, it offers guarantees and subsidies to farmers and technicians. Japan has created special funding policies for the application and cross-discipline of digital agricultural innovation technologies in order to promote the friendliness of these technologies.

(4) Urbanization needs to be vigorously pushed while new digital infrastructure development is encouraged. Urbanization in developing nations can advance the modernization of agriculture, also can support the development of agricultural ecological protection by supplying better environmental and production factor resources. By realizing this win–win situation of integrating ecological civilization into the new urbanization process, digital infrastructure can thus better play its role in advancing AEE.

By serving as a means of connectivity, resource allocation, and scientific control of information technology, agricultural digitization has increased agricultural production efficiency and altered how agriculture is produced and operated, which also has an effect on the ecological environment. These areas of follow-up study can be developed further: (1) Measurements for agricultural digitalization should be more thorough and precise. Construction of infrastructure is a crucial component of agricultural digitization, which may also be assessed from a variety of angles and levels, including the digital environment and digital applications. (2) Does agricultural digitalization affect various agricultural types, such as plantations, cattle, forestry, fisheries, and secondary industries, differently in terms of the environment? For developing nations to adopt sustainable agricultural development plans that take into account regional realities, these research findings have positive policy implications. (3) Future studies may incorporate the network spillover effects of agricultural digitization into the study framework since the influence of agricultural digitization on ecological environment may be able to transcend conventional geographic space and display spatial spillover effects.

## 6. Research Conclusions

In order to explore the impact and conductive mechanism of digitization on agricultural environmental protection, this paper innovatively incorporates new digital infrastructure into the framework of AEE, and further discusses the threshold effect of innovative infrastructure and the moderating role of urbanization. In addition, this paper measures the trend and current status of new digital infrastructure and AEE in China for 30 provinces from 2011 to 2020, and verifies the above mechanisms.

The findings show that: 

(1) New digital infrastructure has a significant contribution to AEE improvement, and this conclusion is still valid through robustness and endogeneity tests.

(2) When dividing new digital infrastructure into three sub-dimensions as information infrastructure, integration infrastructure, and innovation infrastructure, both information infrastructure and integration infrastructure have a significant positive effect on AEE, and information infrastructure is more effective than integration infrastructure. However, unlike the above two sub-dimensions, there is an inverted “U” shaped relationship between innovation infrastructure and AEE, and there is a single threshold between them. When the innovation infrastructure index is lower than 0.2813, innovation infrastructure can significantly promote AEE; when the innovation infrastructure index exceeds 0.2813, innovation infrastructure plays a negative role in it. 

(3) The moderating effect mechanism indicates that the level of urbanization strengthens the promotion effect of new digital infrastructure on AEE.

(4) The heterogeneity test shows that the effect of new digital infrastructure on AEE is stronger in regions with developed traditional transportation facilities and periods when the government pays more attention to agroecological issues.

## Figures and Tables

**Figure 1 ijerph-20-03552-f001:**
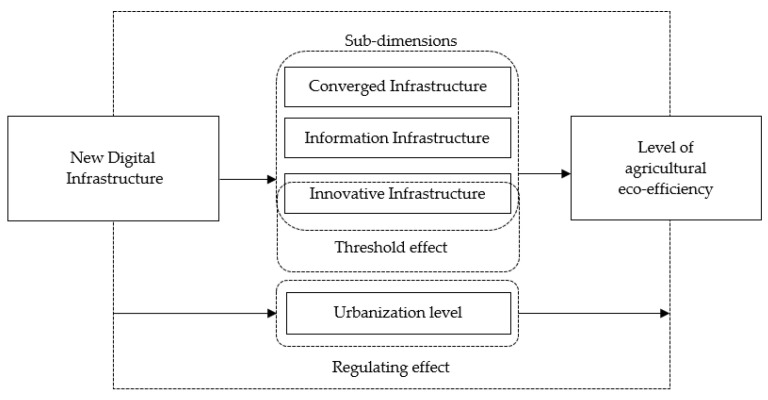
Mechanism of the role of new digital infrastructure on the level of AEE.

**Figure 2 ijerph-20-03552-f002:**
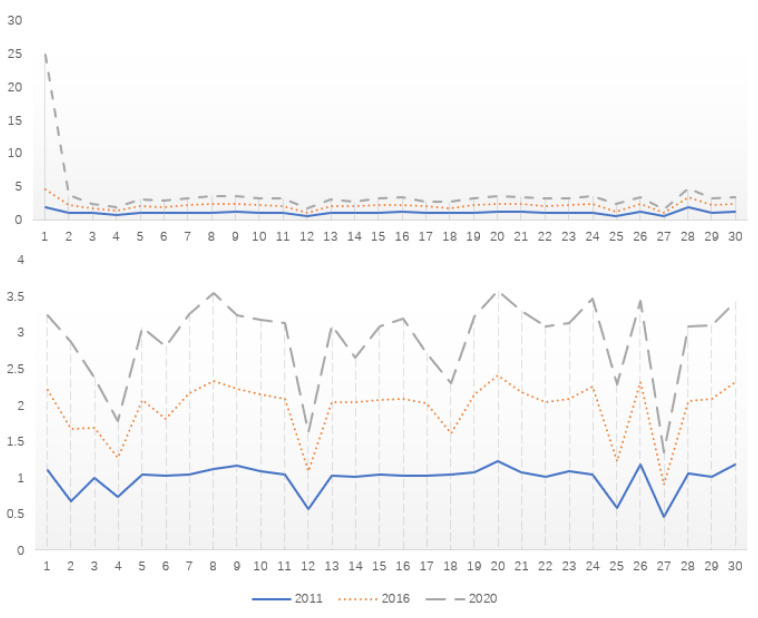
Schematic diagram of changes in AEE levels (**top**: variable returns to scale; **bottom**: constant returns to scale). Note: Horizontal axes 1–30 denote the 30 Chinese provinces of Beijing, Tianjin, Hebei, Shanxi, Inner Mongolia, Liaoning, Jilin, Heilongjiang, Shanghai, Jiangsu, Zhejiang, Anhui, Fujian, Jiangxi, Shandong, Henan, Hubei, Hunan, Guangdong, Guangxi, Hainan, Chongqing, Sichuan, Guizhou, Yunnan, Shaanxi, Gansu, Qinghai, Ningxia, and Xinjiang, in that order.

**Table 1 ijerph-20-03552-t001:** Basis for determining carbon uptake in agriculture.

Crop	Economic Coefficient	Water Content	Uptake Rate	Crop	Economic Coefficient	Water Content	Uptake Rate
Rice	0.45	12%	0.414	Potato	0.70	70%	0.423
Wheat	0.40	12%	0.485	Sugar cane	0.50	50%	0.450
Corn	0.40	13%	0.471	Sugar beets	0.70	75%	0.407
Beans	0.34	13%	0.450	Vegetables	0.60	90%	0.450
Canola	0.25	10%	0.450	Melon	0.70	90%	0.450
Peanut	0.43	10%	0.450	Tobacco	0.55	85%	0.450
Sunflower	0.30	10%	0.450	Other crops	0.40	12%	0.450
Cotton	0.10	8%	0.450				

**Table 2 ijerph-20-03552-t002:** Basis for determining carbon emission coefficients for agriculture.

Carbon Emission Sources	Coefficients	Reference Value Sources
Pesticides	4.9341 kg/kg	Oak Ridge National Laboratory (USA)
Fertilizer	0.8956 kg/kg	Oak Ridge National Laboratory (USA)
Diesel	0.5927 kg/kg	IPCC
Agricultural film	5.18 kg/kg	Institute of Agricultural Resources and Ecological Environment, Nanjing Agricultural University
Irrigation	266.48 kg/hm^2^	Duan et al. [52]
Tillage	312.6 kg/hm^2^	Wu et al. [53]

**Table 3 ijerph-20-03552-t003:** New digital infrastructure indicator system.

Primary Indicators		Secondary Indicators	Units	Indicator Direction
Information Infrastructure		Local exchange capacity	10,000 doors	Positive
		Mobile Phone Switching Capacity	million homes	Positive
		Cell phone base station	million	Positive
		Fiber optic cable line length	meters	Positive
		Number of domain names	million	Positive
		Number of websites	million	Positive
		Number of IPv4 addresses	million	Positive
		Internet broadband access ports	million	Positive
		Software business revenue	million	Positive
		Number of computers in use at the end of the period	table	Positive
		Computing base per 100 people	table	Positive
		Number of websites owned by enterprises	individual	Positive
		Number of websites owned by each 100 enterprises	individual	Positive
		E-commerce sales	billion yuan	Positive
Convergence Infrastructure	Traditional Infrastructure	Railroad mileage	Kilometer	Positive
		High-speed grade highway mileage	kilometer	Positive
		Bus and trolley bus mileage	kilometer	Positive
		Rail mileage	kilometer	Positive
	Informatization degree	Software business revenue	million yuan	Positive
		Number of computers in use at the end of the period	table	Positive
		Number of computers per 100 people	table	Positive
		Number of websites owned by enterprises	individual	Positive
		Number of websites owned by each 100 enterprises	individual	Positive
		E-commerce sales	billion yuan	Positive
Innovation Infrastructure		R&D investment intensity	%	Positive
		Number of institutions	individual	Positive
		Total R&D personnel	people	Positive
		R&D personnel equivalent full time	per person per year	Positive
		R&D funding internal expenditure	million	Positive
		Government Funds	million yuan	Positive
		R&D subject input funds	ten thousand yuan	Positive
		Number of patent applications	piece	Positive

**Table 4 ijerph-20-03552-t004:** Level of new digital infrastructure.

Province	2011	2020	Province	2011	2020
Beijing	0.7480	0.7192	Henan	0.2543	0.3077
Tianjin	0.2362	0.2511	Hubei	0.2751	0.3381
Hebei	0.2392	0.2944	Hunan	0.2178	0.2826
Shanxi	0.1349	0.1609	Guangdong	0.9178	0.9559
Inner Mongolia	0.1278	0.1793	Guangxi	0.1643	0.1715
Liaoning	0.3319	0.2847	Hainan	0.0920	0.0908
Ji Lin	0.1421	0.1620	Chongqing	0.2029	0.2582
Heilongjiang	0.1438	0.1899	Sichuan	0.3174	0.4160
Shanghai	0.5478	0.5286	Guizhou	0.1220	0.1589
Jiangsu	0.7708	0.6788	Yunnan	0.1677	0.2070
Zhejiang	0.5339	0.5769	Shaanxi	0.2142	0.2845
Anhui	0.2222	0.3102	Gansu	0.1153	0.1186
Fujian	0.3016	0.3026	Qinghai	0.0632	0.0980
Jiangxi	0.1451	0.2318	Ningxia	0.0660	0.0674
Shandong	0.4362	0.5094	Xinjiang	0.1097	0.1063

**Table 5 ijerph-20-03552-t005:** Descriptive statistics.

Variable	Codes	Sample Size	Mean Value	Standard Deviation	Maximum Value	Minimum Value
New Digital Infrastructure	NDI	300	0.302	0.211	0.0632	0.956
Converged Infrastructure	CI	300	0.406	0.226	0.0746	1
Innovation Infrastructure	INNI	300	0.214	0.150	0.0726	0.801
Information Infrastructure	INFI	300	0.256	0.177	0.0726	0.875
AEE with Constant Returns to Scale	CRS-AEE	300	0.966	0.217	0.401	1.279
AEE with Variable Returns to Scale	VRS-AEE	300	1.140	1.176	0.421	20.29
Level of Agricultural Economic Development	AED	300	4.692	5.389	0.146	76.95
Planting Structure	PS	300	0.649	0.140	0.355	0.971
Agricultural Disaster Rate	ADR	300	0.151	0.119	0	0.619
Rural Human Capital	RHC	300	2.044	0.0778	1.766	2.268
Financial Support to Agriculture	FSA	300	0.115	0.0336	0.0374	0.217
Level of Urbanization	LU	300	58.14	12.05	34.96	89.60

**Table 6 ijerph-20-03552-t006:** Baseline regression results.

	VRS-AEE				CRS-AEE			
	(1)	(2)	(3)	(4)	(5)	(6)	(7)	(8)
NDI	1.0237 **				0.1917 **			
	(2.41)				(2.56)			
INFI		1.4666 ***				0.2169 **		
		(3.09)				(2.58)		
CI			0.9352 **				0.1964 ***	
			(2.24)				(2.68)	
INNI				−0.0168				−0.3417 ***
				(−0.03)				(−4.16)
AED	−0.0033	−0.0034	−0.0037	−0.0029	0.0036	0.0037	0.0035	0.0035
	(−0.26)	(−0.27)	(−0.29)	(−0.22)	(1.62)	(1.63)	(1.58)	(1.58)
ADR	−0.4523	−0.4329	−0.4449	−0.6034	−0.3342 ***	−0.3373 ***	−0.3291 ***	−0.3301 ***
	(−0.76)	(−0.73)	(−0.75)	(−1.01)	(−3.19)	(−3.23)	(−3.14)	(−3.22)
PS	−0.9796 **	−0.8642 *	−1.0386 **	−1.0780 **	−0.3335 ***	−0.3204 ***	−0.3436 ***	−0.2724 ***
	(−2.03)	(−1.79)	(−2.15)	(−2.16)	(−3.92)	(−3.75)	(−4.06)	(−3.19)
RHC	0.9122	0.8185	1.0057	1.0943	0.4055 ***	0.3989 ***	0.4209 ***	0.3721 **
	(1.05)	(0.95)	(1.16)	(1.25)	(2.66)	(2.61)	(2.77)	(2.48)
FSA	1.3899	1.7025	1.6100	−2.6385	0.9668 **	0.8555 *	1.1043 **	−0.2209
	(0.51)	(0.67)	(0.56)	(−1.17)	(2.02)	(1.89)	(2.20)	(−0.57)
N	300	300	300	300	300	300	300	300

Note: ① *, **, *** indicate significant at the 10%, 5%, and 1% levels, respectively; ② t-values in parentheses.

**Table 7 ijerph-20-03552-t007:** Test of threshold effect of innovation infrastructure.

Type	Threshold	F Value	*p* Value	Sampling Number
Single Threshold	0.2813	9.28	0.0233	300
Double Threshold	0.3169	4.20	1.0000	300
	0.3233			

**Table 8 ijerph-20-03552-t008:** Innovative infrastructure and VRS AEE threshold value.

VRS-AEE	
INNI	14.9609 ***
$\left({\mathrm{INNI}}_{\mathrm{it}}\leq 0.2813\right)$	(6.21)
INNI	−4.1182 **
$\left({\mathrm{INNI}}_{\mathrm{it}}>0.2813\right)$	(−2.36)
Control variables	Yes
N	300

Note: ① **, *** indicate significant at the 5%, and 1% levels, respectively; ② t-values in parentheses.

**Table 9 ijerph-20-03552-t009:** Robustness test.

	VRS-AEE	CRS-AEE	VRS-AEE	CRS-AEE
	(1)	(2)	(3)	(4)
NDI			1.0135 **	0.1933 **
			(2.40)	(2.55)
L.NDI	1.1286 **	0.1917 **		
	(2.39)	(2.56)		
Control variables	Yes	Yes	Yes	Yes
N	270	270	300	300

Note: ① ** indicate significant at the 5% levels; ② t-values in parentheses.

**Table 10 ijerph-20-03552-t010:** Endogeneity test.

	First Stage	Second Stage	
		VRS-AEE	CRS-AEE
NDI		2.6976 ***	
		(3.72)	
IV	0.0147 ***		0.3865 ***
	(12.64)		(3.07)
Control variables	Yes	Yes	Yes
N	300	300	300
F value	170.777		

Note: ① *** indicate significant at the 1% levels; ② Here the first stage brackets indicate the t value and the second stage brackets indicate the Z value.

**Table 11 ijerph-20-03552-t011:** Heterogeneity analysis.

	Government Attention		Degree of Development of Traditional Transportation Facilities	
	2011–2016	2017–2020	Underdeveloped	Developed
	(1)	(2)	(3)	(4)
NDI	0.2262 *	2.6258 **	0.9427	0.2753 **
	(2.34)	(1.70)	(1.30)	(2.06)
Control variables	Yes	Yes	Yes	Yes
N	180	120	150	150

Note: ① *, ** indicate significant at the 10% and 5% levels, respectively; ② t-values in parentheses.

**Table 12 ijerph-20-03552-t012:** Interaction effect of urbanization level and new digital infrastructure (explained AEE).

	VRS-AEE	CRS-AEE
	(1)	(2)
NDI	−9.7882 ***	−0.0530
	(−4.65)	(−0.14)
LU	−0.0278 **	0.0020
	(−2.08)	(0.79)
NDI×LU	0.1580 ***	0.0029
	(5.00)	(0.50)
Control variables	Yes	Yes
N	300	300

Note: ① **, *** indicate significant at the 5% and 1% levels, respectively; ② t-values in parentheses.

## Data Availability

All sample data sets are downloaded from the website. Data are available at http://www.stats.gov.cn/ (accessed on 31 December 2021).

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
