# Peer review of "New Digital Infrastructure’s Impact on Agricultural Eco-Efficiency Improvement: Influence Mechanism and Empirical Test—Evidence from China"

_ijerph, 2023, doi:10.3390/ijerph20043552_

Round 1

Reviewer 1 Report

This paper innovatively incorporates new digital infrastructure and urbanization level into of the concept of agricultural eco-efficiency (AEE) and adopts SBM-DEA model, entropy weighting method, and mixed regression to analyze, based on the sample data of the 30 provinces of China from 2011 to 2020. The above results also provide rich insights for China and other similar developing countries on how to balance the agriculture digitization and AEE.

After minor revision, it is recommended to publish the paper.

It is recommended that the paper can be improved in the following points.

1. Please explain more about why AEE is used, and what are the benefits to use this.

2. In 3.2.1, the paper has measured AEE in China and generally describe the variation trend. Please provide a more thorough explanation of the characteristics and causes of this AEE transformation in the context of Chinese agriculture's economic and technological development.

3. The policy recommendation section needs to strengthen the insight into the governance of AEE in developing countries.

Author Response

The authors wish to thank the reviewer for the very helpful comments and suggestions. The manuscript has been revised according to the reviewer's suggestions and all changes have been highlighted in red in the revised manuscript. We have addressed the comments raised by the reviewer and listed the responses and modifications point by point below. 

 Point 1: Please explain more about why AEE is used, and what are the benefits to use this.

 Response 1: By contrasting AEE with the idea of conventional agriculture, we explain in paragraph 3 why it is more appropriate to employ AEE as an explanatory variable in this study due to its benefits in balancing agricultural inputs, outputs, and ecological benefits. More can be seen in the red part of INTRODUCTION.

Point 2: In 3.2.1, the paper has measured AEE in China and generally describe the variation trend. Please provide a more thorough explanation of the characteristics and causes of this AEE transformation in the context of Chinese agriculture's economic and technological development.

Response 2: In 3.2.1, below Figure 1, the resource endowment, digital base, agricultural features, and technical progress in the eastern and western regions of China are further used to explain the variances and variations in AEE across Chinese provinces during the sample period. More can be seen in the red part of 3.2.1.

Point 3: The policy recommendation section needs to strengthen the insight into the governance of AEE in developing countries.

Response 3: In the fifth part "Research conclusions, policy recommendations and research outlook", policy recommendations are reinforced from the perspective of developing countries in four areas: multi-actor investment in new digital infrastructure, convergence value creation, low-cost inclusive technology innovation, and urbanization. More can be seen in the red part of RESEARCH CONCLUSION, POLICY RECOMMENDATIONS AND RESEARCH OUTLOOK.

Reviewer 2 Report

In In order to explore the impact and conductive mechanism of digitization on agricultural environmental protection, this paper innovatively incorporates new digital infrastructure into the framework of AEE, and further discusses the threshold effect of innovative infrastructure and the moderating role of urbanization. In addition, this paper measures the trend and current status of new digital infrastructure and AEE in China for 30 provinces from 2011 to 2020, and verifies the above mechanisms. After necessary revision, it is recommended to publish the paper.The following modifications are suggested:

 1. Relevant research background needs to be supplemented in INTRODUCTION.

 2. The interpretation of the empirical part suggests a more in-depth and detailed analysis in the context of the application of digital technologies in sustainable agriculture in China, and highlights the role of new digital infrastructure in environmental protection and public health.

 3. The conclusion should correspond to the previous section and present actionable guidelines.

Author Response

The authors wish to thank the reviewer for the very helpful comments and suggestions. The manuscript has been revised according to the reviewer's suggestions and all changes have been highlighted in red in the revised manuscript. We have addressed the comments raised by the reviewer and listed the responses and modifications point by point below.

Point 1: Relevant research background needs to be supplemented in INTRODUCTION.

Response 1: In the third paragraph of INTRODUCTION, literature discussing the factors influencing AEE has been added. The majority of the research now has focused more on climatic, agricultural, social, and economic aspects than on agricultural digitization, and even less on the dual effects of digitization on agroecology. More can be seen in the red part of INTRODUCTION.

Point 2: The interpretation of the empirical part suggests a more in-depth and detailed analysis in the context of the application of digital technologies in sustainable agriculture in China, and highlights the role of new digital infrastructure in environmental protection and public health.

Response 2: In the last paragraph of 4.1, we add the explanation of the empirical results from the perspective of information infrastructure and convergence infrastructure. More can be seen in the red part of 4.1.

 Point 3: The conclusion should correspond to the previous section and present actionable guidelines.

Response 3: In light of the prior discussion, the conclusion and policy recommendations have been strengthened, and further operational recommendations have been provided.. More can be seen in the red part of RESEARCH CONCLUSION, POLICY RECOMMENDATIONS AND RESEARCH OUTLOOK.

Reviewer 3 Report

The manuscript "New Digital Infrastructure's Impact on Agricultural Eco-efficiency Improvement: Influence Mechanism and Empirical Test - Evidence from China" seems interesting. New agricultural digital infrastructure can boost eco-efficiency. Resource management precision, industrial process optimization, and environmental performance monitoring and assessment are the influencing mechanisms. Digital infrastructure increases efficiency and reduces waste in agriculture, according to Chinese studies. Digital infrastructure can improve agricultural eco-efficiency, but government regulations, technological diffusion, and farmer adoption affect its performance.

Although, It's well-written. Before accepting the paper, here are some general points that are needed to address and included in the appropriate section of the manuscript.

#Are significant points mentioned in the MS relevant to current field developments? 

#Please offer a complete and up-to-date literature study on the topic? 

#Does the study's methodology fit the research question? Are the procedures detailed enough to reproduce the study? Please discuss in the relevant section.

#The study results clear, accurate, and well-presented. The should support the paper's conclusions.

#Does the paper address study limitations? Are limitations balanced and critical? It should be discussed in the manuscript.

It should be addressed...................Does the paper advance the field? Is the paper informative?

Author Response

The authors wish to thank the reviewer for the very helpful comments and suggestions. The manuscript has been revised according to the reviewer's suggestions and all changes have been highlighted in red in the revised manuscript. We have addressed the comments raised by the reviewer and listed the responses and modifications point by point below.

Point 1: Are significant points mentioned in the MS relevant to current field developments? Please offer a complete and up-to-date literature study on the topic?

Response 1: In the third paragraph of INTRODUCTION, literature discussing the factors influencing AEE has been added. We also add up-to-date literature study here. The majority of the research now has focused more on climatic, agricultural, social, and economic aspects than on agricultural digitization, and even less on the dual effects of digitization on agroecology. More can be seen in the red part of INTRODUCTION.

 Point 2: Does the study's methodology fit the research question? Are the procedures detailed enough to reproduce the study? Please discuss in the relevant section. The study results clear, accurate, and well-presented. The should support the paper's conclusions. 

 Response 2: The study’s methodology fit the research question, which has been explained in 3.1. And in the last paragraph of 4.1, we add the explanation of the empirical results from the perspective of information infrastructure and convergence infrastructure. More can be seen in the red part of 4.1. 

 Point 3: Does the paper address study limitations? Are limitations balanced and critical? It should be discussed in the manuscript. It should be addressed that the paper advance the field? Is the paper informative? 

 Response 3: We go into further detail about this paper's shortcomings and potential directions for future research in the article's final paragraph. More can be seen in the red part of last paragraph.

Round 2

Reviewer 3 Report

The authors have revised MS as per the suggestions and now can be accepted in its present form.